# A phenomenological comparison of the effects of blue light, red light and radio waves on the escape speed of *Caenorhabditis elegans* and the rate of closure of *Gerbera jamesonii* petals

Alexander W. Kline[1☯], Charles S. Beattie[1☯], Addison K. Shenk[2‡], Samuel I. Spicher[2‡], Timothy A. Bloss[3], Laura Tipton[3,4], Marquis T. Walker[3], Laura G. Vallier[5], Kristopher L. Schmidt[2]*, Giovanna Scarel[1]*

1 Department of Physics and Astronomy, James Madison University, Harrisonburg, Virginia, United States of America, 2 Department of Biology and Chemistry, Eastern Mennonite University, Harrisonburg, Virginia, United States of America, 3 Department of Biology, James Madison University, Harrisonburg, Virginia, United States of America, 4 Department of Mathematics & Statistics, James Madison University, Harrisonburg, Virginia, United States of America, 5 Department of Biology, Hofstra University, Hempstead, New York, United States of America

☯ These authors contributed equally to this work
‡ These authors also contributed equally to this work
* scarelgx@jmu.edu (GS), kristopher.schmidt@emu.edu (KS)

## Abstract

### Objective

Great interest surrounds understanding the effects of radio waves on biological organisms, including humans, animals and plants. Several prior studies, however, showed contradictory results. We hypothesized that the problem lay in the lack of a method for evaluating the energy transferred from radio waves, or electromagnetic waves in general, to biological organisms. Therefore, we proposed to measure the transferred energy with the classic electromagnetic wave energy, i.e., the product of the intensity and the inverse of the frequency of the waves.

### Methods

To test this hypothesis, we exposed two simple light-sensitive biological organisms, *Caenorhabditis elegans* (*C. elegans*) and *Gerbera jamesonii* (*G. jamesonii*), to radio waves, red light, blue light and white light. We selected frequency and intensity such that each type of electromagnetic wave could transfer a similar amount of energy to the biological organisms. We then observed the kinematic and postural response of *C. elegans*, and the rate of closure of *G. jamesonii*'s petals to assess whether similar effects would be detected when the energy at different frequencies is similar.

**Data availability statement:** All relevant data are within the manuscript and its Supporting Information files.

**Funding:** This study was financially supported by the National Science Foundation in the form of a Major Research Instrumentation grant awarded to KS (1827997). This study was also financially supported by the 4-VA at James Madison University (JMU) in the form of a grant awarded to GS, KS, MW, and TB. This study received further financial support from the JMU Department of Physics and Astronomy in the form of summer salary awarded to AK to support their work on this project. Additional financial support was provided by the Jeffrey E. Tickle '90 Family Endowment in Science & Mathematics Scholarship to fund the summer salary of CB and support their work on this project. Financial support was also provided by the Eastern Mennonite University Department of Biology & Chemistry in the form of summer salaries for AS and SS to support their work on this project. This study was also supported by the JMU Department of Biology in the form of administrative assistance in managing the 4-VA JMU grant. The funders had no role in study design, data collection and analysis, decision to publish, or preparation of the manuscript.

**Competing interests:** the authors have NO competing interests to declare.

## Conclusions

In both *C. elegans* and *G. jamesonii*, we found that radio waves trigger effects like those generated by light with similar energy. This outcome provides support to our hypothesis. We therefore infer that electromagnetic wave intensity needs to be considered when estimating possible harm linked to the exposure of biological organisms to radio waves or, in general, electromagnetic waves. Moreover, the successful ability of radio waves in biological organisms to produce effects like those produced by blue light, together with their long penetration depth in tissues, stimulates the investigation of radio waves as a substitute for blue light in a non-invasive version of optogenetics.

## Introduction

Radio waves and microwaves are electromagnetic waves with frequency ($\nu$) in the range between 50 MHz and 300 GHz (or wavelength λ between 6 m and 1 mm). When they interact with a receptor, for example in a biological organism (humans, animals and plants), radio waves and microwaves transfer their energy thus activating processes and reactions that can be both beneficial and harmful. Since radio waves and microwaves are ubiquitously present in the Earth's atmosphere due to radio- and telecommunications, many studies have tried to establish their effects on biological organisms. Often, however, attention focused on the actual effects, neglecting the magnitude of the wave's energy and intensity, and thus preventing establishment of a threshold between safe and harmful levels of exposure.

In the literature, prior studies related the energy-intensity values of radio waves and microwaves to their biological effects through various routes. Some authors focused on frequency [1,2], others on intensity [3] and exposure duration [4]. Bertil Persson *et al.* [1] and Catrin Bauréus Koch *et al.* [2] found that electromagnetic waves with very low frequency affect $Ca^{2+}$ influx and efflux in the common spinach plant *Spinacia oleracea L*. These researchers however did not provide precise information on energy and intensity. Li Zhao *et al.* [3] found that microwaves affect cellular activity and apoptosis in hippocampal neurons, and that different intensities influence these effects. Unfortunately, these authors used microwaves in a very limited frequency and intensity range ($\nu$ from 120 GHz to 157 GHz, and intensity from 10 mW to 50 mW). Finally, Yan Gao *et al.* [3] showed that an exposure of 60 hours of the nematode *Caenorhabditis elegans* (*C. elegans*) to microwaves at $\nu = 1.75$ GHz and intensity up to 3 nW induced variations in gene expression. However, neither Li Zhao *et al*. [3] nor Yan Gao *et al.* [4] justified why they selected intensities in the mW and nW range, respectively. Currently, literature does not offer criteria on how to choose the magnitude of the intensity of radio waves and microwaves and how intensity, frequency, and duration of exposure are related to the energy transferred from the waves to a biological organism. Without this information no threshold can be established between safe and harmful radio wave and microwave energies.

In this work we investigated the relationship between intensity, frequency and energy of radio waves which could lead to establishing (1) a procedure to estimate

the energy threshold above which electromagnetic waves are dangerous and (2) a criterion to select the intensity range where electromagnetic waves carry an energy below that threshold. In our experiments we directed either light or radio waves toward two biological organisms known to be sensitive to electromagnetic waves in the form of visible light: *Caenorhabditis elegans* (*C. elegans*) and *Gerbera jamesonii* (*G. jamesonii*). To proceed, we considered the recent proposal that attributed to periodic phenomena, which include electromagnetic waves, the classic energy consisting of the product of the intensity P and the inverse of the frequency $\nu$, or $\frac{P}{\nu}$ [5,6]. We reasoned that, by properly selecting P and $\nu$, radio waves supply the same amount of energy as visible light. Thus, we investigated the intensities enabling blue light, red light and radio waves to transfer the same amount of energy and generate similar responses from *C. elegans* and *G. jamesonii*. We concluded that the classic energy $\frac{P}{\nu}$ accounts for the energy of electromagnetic waves transferred to biological organisms and thus can be used to establish a criterion to determine the intensity range in which the energy of electromagnetic waves of a certain frequency is dangerous.

## Materials and methods

### C. elegans

*C. elegans* are non-pathogenic soil-dwelling nematodes that are widely employed in research as a genetic model organism for the nervous and other systems, and to observe the process of evolution at an accelerated pace. These nematodes are light-sensitive: blue light, for instance, is detrimental to *C. elegans* longevity and triggers a strong avoidance reaction [7,8].

For our experiments, populations of *C. elegans* wild-type N2 were maintained under standard laboratory conditions at 22°C on nematode growth medium (NGM) seeded with *Escherichia coli* (*E. coli*) OP50 [9]. To standardize the developmental stages for the experiments, young adult hermaphrodites were synchronized using a hypochlorite treatment [10]. For each of the two-three experimental trials, at least ten worms were transferred to each 60 mm NGM supplemented with 25 μL of *E. coli* OP50, prepared from a fresh overnight culture (N = 40). To constrain worm movement during exposure to light or radio waves and video recording, a copper ring was placed around the bacterial lawn prior to worm transfer.

### G. jamesonii

*G. jamesonii* are flowering plants readily available, easy to maintain and observe. Their flowers exhibit sharp nyctinastic movements observable without the aid of a microscope and in which the petals open at sunrise and close at sunset. Multi-color plants of *G. jamesonii* were purchased from Monrovia Inc., used as received, and observed *in vivo*. Flower color was not counted as a variable in the experiments. As-received plants were acclimatized three to four days before blooming in their illumination environment and were never removed from that area during the experiment. Each flower selected for an experimental trial was used once.

### Design of the experiments with C. elegans

Light, especially blue light, elicits an avoidance reaction in *C. elegans*. We assumed this phenomenon to be caused by the amount of energy transferred from light to nematodes. We then supposed that radio waves capable of transferring the same amount of energy as light should also trigger an avoidance reaction. In this work we tested this hypothesis.

### Design of the experiments with G. jamesonii

Earth's motion around its axis gives rise to the daily cycle of bright and dark periods related to exposure to solar white light or lack thereof. This phenomenon gives rise to circadian cycles in biological organisms. One of the consequences in *G. jamesonii* is the daily opening and closing of the petals, known as nyctinastic movement. Petal closure in *G. jamesonii* is especially evident in the time interval between 13:30 and 16:30 of days from mid-May to mid-August. We hypothesized

that nyctinastic movements depend upon the cyclical supply of energy transferred from light to petals. We reasoned, therefore, that persistent darkness should inhibit nyctinastic movements. We also argued that, due to the lack of periodic exposure and non-exposure to light or other electromagnetic waves, continuous exposure to light at a constant intensity should inhibit nyctinastic movements, perhaps to a lesser degree than darkness, because continuous exposure to light was reported to elicit a wavelength-dependent activation of phytohormones [11]. On the other hand, we thought that if radio waves transfer less energy than light then they would inhibit nyctinastic movements. However, if radio waves can transfer the same amount of energy as light, then they too should only partially inhibit nyctinastic movements on *G. jamesonii* petals continuously exposed to them. We tested this hypothesis by comparing the effects of continuous exposure of *G. jamesonii* to dark, blue light, red light, and radio waves. We exposed the flowers to solar white light to directly observe sharp nyctinastic movements and evaluate to what degree continuous exposure to electromagnetic waves suppresses or inhibits them. The results we obtained with flowers in dark acted as controls, as they represented the environment without electromagnetic waves.

### Temperature measurement of *C. elegans* and *G. jamesonii*

To establish whether heat or electromagnetic wave energy determined the response of nematodes, we measured *C. elegans* temperature with a Govee WiFi Thermometer Hygrometer H5103. For *G. jamesonii*, to clarify whether relevant temperature changes were present during illumination, we used six Cole-Palmer Digi-Sense General Purpose Liquid-In-Glass Thermometers placed in each of the environments in which we performed the experiments.

### Experimental set up

To minimize disturbances, we performed our experiments after placing either *C. elegans* or *G. jamesonii* and the electromagnetic wave sources in an expanded polystyrene (EPS) enclosure located in a windowless laboratory kept at 18°C-20°C and with lights off. Without shielding the radio waves are dispersed and do not act efficiently. We draw this conclusion from measurements we performed with dedicated detectors and reported in section 1 of S1 Text. In the experiments, *C. elegans* plates were placed under the microscope which was used for video recording inside the EPS enclosure. In the experiments with *G. jamesonii*, the EPS enclosures were opened only for 1–2 minutes, the time interval necessary for photo recording. During those 1–2 minutes we kept the laboratory lights on and turned them again off immediately after. The flowers examined under exposure to solar white light were not kept in an EPS enclosure but on a windowsill facing East (87° E measured with a Melon Soft Compass App) of a room kept at 17.5°C–18.5°C and illuminated by sunlight in the morning. The photographs of *G. jamesonii* in solar white light were taken without room's fluorescent lights on.

### Electromagnetic wave sources

We used blue light and red light from light emitting diodes (LEDs) (ALLECIN-3 mm). Light emitting diodes are recognized as biologically meaningful light sources for nematode behavioral experiments [12]. We obtained radio waves from four radio devices (Sony ICF38 portable AM/FM) [6]. To avoid any influence of sound [13], we fixed the volume of the four radio devices to zero. In experiments with *C. elegans*, white light was obtained from a Thorlabs SLS204 lamp and used at 25% of the maximum allowed intensity. To assess the effect of temperature, we compared the outcome of the exposure to white light at 25% and 100% of the maximum allowed intensity. In experiments with *G. jamesonii* white light was the natural solar white light.

### Intensity and energy of light and radio waves

To determine intensity and energy of blue light, red light, and radio waves, we used the following protocol. We first measured the intensity (P) of blue light, red light, and radio waves according to the procedure outlined by R. Joseph

Rybarzyck *et al.* [6] using a detector (3TECBT) described by those authors. Next, we estimated the specific intensity, which we called P*, on our biological organisms (*C. elegans* or *G. jamesonii*). We did so by multiplying P to the ratio between the size d of either *C. elegans* (1 mm) or *G. jamesonii* (3 cm on average) and the size D of the 3TECBT detector (2.792 cm). This geometry-related ratio (P$\frac{d}{D}$) represents the *antenna factor*. We presented the values of P* of blue light, red light and radio waves as used in our experiments in Table 1 for *C. elegans* and Table 2 for *G. jamesonii*. In these tables we also reported the classic electromagnetic wave energies P*/$\nu$ [5,6] of blue light, red light and radio waves. Finally, assuming a surface area of about 1 mm² for *C. elegans* and 6 cm² for the petals of *G. jamesonii*, we estimated the intensities I in W/m² and reported them in Tables 1 and 2. Since white light is a mixture of different frequencies, it was not possible to associate P* with a unique frequency and obtain unique energy. Therefore, for white light supplied either by the Thorlabs SLS204 lamp in the experiments with *C. elegans* or by solar white light in experiments with *G. jamesonii* we did not collect information about intensity.

**The classic electromagnetic wave energy**

To determine the energy transferred from blue light, red light, and radio waves we used the classic electromagnetic wave energy P*/$\nu$ reported and discussed in Refs. 5 and 6 for photodetectors and, generally, for periodic phenomena, respectively. In this section we relate the classic electromagnetic wave energy to Maxwell's equations and show its effectiveness, respectively, in two cases: a) electromagnetic waves emitted by a simple oscillating dipole [14,15] and b) electromagnetic waves absorbed by rotating nanoparticles [16,17]. More details on the derivation of Larmor's Equation from Maxwell's equations in classical electrodynamics and other examples can be found in section 2 of S1 Text and in a GitHub repository at https://github.com/Charlesbt12/Supplementary-1-Maxwell-to-Larmor-PLOS-one-350.

**Case a)** The power $P_{Larmor}$ emitted by a simple oscillating dipole is given by Larmor's equation [14,15]. Considering $p_0 = qd_{dip}$ the dipole's moment, q the magnitude of the charges in the dipole, $d_{dip}$ their distance, $\varepsilon_0 = 8.854 \times 10^{-12} \frac{F}{m}$ the dielectric permittivity in vacuum, $\nu$ and $\lambda$ the frequency and wavelength, respectively, of the emitted electromagnetic waves, then $P_{Larmor}$ is:

$$P_{Larmor} = \frac{4}{3}\pi^3 \frac{p_0^2}{\varepsilon_0} \frac{\nu}{\lambda^3}$$

(1)

**Table 1. Characteristics of the electromagnetic waves used in the experiments with *C. elegans*.**

| EM wave | λ | $\nu$ | τ | P* | P*/$\nu$ | I |
|---------|-----|-----|-----|-----|-----|-----|
| Radio | 3 m | 100 MHz | 10 ns | 0.34 nW | 3.4 aJ | 0.34 mW/m² |
| Red | 700 nm | 0.43 PHz | 2.33 fs | 0.14 mW | 0.3 aJ | 0.14 mW/m² |
| Blue | 450 nm | 0.67 PHz | 1.5 fs | 1.70 mW | 2.5 aJ | 1.7 mW/m² |

In the experiments with *C. elegans* we used the listed wavelength λ, frequency $\nu$, period τ, intensity P*, energy P*/$\nu$, and intensity in W/m² of radio waves, red and blue light.

**Table 2. Characteristics of the electromagnetic waves used in the experiments with *G. jamesonii*.**

| EM wave | λ | $\nu$ | τ | P* | P*/$\nu$ | I |
|---------|-----|-----|-----|-----|-----|-----|
| Radio | 3 m | 100 MHz | 10 ns | 9.5 nW | 95 aJ | 95 µW/m² |
| Red | 700 nm | 0.43 PHz | 2.33 fs | 3.90 mW | 9.09 aJ | 9.09 µW/m² |
| Blue | 450 nm | 0.67 PHz | 1.5 fs | 12.0 mW | 18.0 aJ | 18.0 µW/m² |

In the experiments with *G. jamesonii* we used the listed wavelength λ, frequency $\nu$, period τ, intensity P*, energy P*/$\nu$, and intensity in W/m² of radio waves, red and blue light.

from which it is straightforward to derive the classic electromagnetic wave energy as:

$$\frac{P_{Larmor}}{\nu} = \frac{4}{3}\pi^3\frac{p_0^2}{\varepsilon_0}\frac{1}{\lambda^3} = \frac{4}{3}\pi^3\frac{q^2 d_{dip}^2}{\varepsilon_0}\frac{1}{\lambda^3}.$$
(2)

Notably, $P_{Larmor}$ has units of $W=\frac{J}{s}$, while $\frac{P_{Larmor}}{\nu}$ has units of J. In classical and relativistic electrodynamics, Eq. (1) is directly derived from Maxwell's equations, as outlined in the textbooks by David Griffiths [14] and John Jackson [15]. More details are provided in section 2 of S1 Text. The presence of $\frac{d_{dip}^2}{\lambda^3}$ in Eq. (2) highlights a geometric factor, referred to as the *antenna factor*.

**Case b)** The energy $E_{rot}$ absorbed by a nanoparticle rotating at a frequency $f_{rot}$ under the illumination of a laser of power $P_{laser}$ and frequency $\nu_{laser}$ is:

$$E_{rot} = \frac{1}{2}I_{nanopar}f_{rot}^2\Gamma_{geom+opt} = \frac{P_{laser}}{\nu_{laser}} + \Gamma_{environ}f_{rot}p_{gas}$$
(3)

In Eq. (3) various nanoparticle's parameters appear, which are defined as follows: $I_{nanopar}$ is the moment of inertia, $f_{rot}$ the rotational frequency, $\Gamma_{geom+opt}$ a dimensionless factor collecting various optical and geometrical characteristics, $\Gamma_{environ}$ a factor determined by the gaseous environment, and $p_{gas}$ is gas pressure. The details are presented in Refs. 16 and 17, and in section 2 of S1 Text. In Eq. (4) below we highlight the relationship between the electromagnetic wave energy $\frac{P_{laser}}{\nu_{laser}}$, the rotational energy $E_{rot}$, and $\Gamma_{environ}f_{rot}p_{gas}$:

$$\frac{P_{laser}}{\nu_{laser}} = \frac{1}{\Gamma_{geom+opt}}\frac{1}{2}I_{nanopar}f_{rot}^2 - \frac{\Gamma_{environ}}{\Gamma_{geom+opt}}f_{rot}p_{gas}.$$
(4)

Equation (4) accounts for the experimental results as summarized in section 2 of S1 Text.

The classic electromagnetic wave energy is characterized by two variables: the electromagnetic wave's power P and the frequency ν. Therefore, properly selecting P enables us to obtain the same energy with different frequencies. For example, green light has $\nu_{green} = 0.564\ 10^{15}$ Hz (corresponding to wavelength $\lambda_{green} = 532$ nm). Selecting $P_{green} = 10\ 10^{-3}$ W (typical power for laser pointers and LEDs), the energy is $\frac{P_{green}}{\nu_{green}} = 17.7\ 10^{-18}$ J. The same energy is achieved by radio waves at $\nu_{radio} = 100$ MHz (or $100\ 10^6$ Hz) with $P_{radio} = 1.77\ 10^{-9}$ W: $\frac{P_{radio}}{\nu_{radio}} = \frac{1.77\ 10^{-9}\ W}{100\ 10^6\ \frac{1}{s}} = 1.77\ 10^{-17}$ J $= 17.7\ 10^{-18}$ J. The power transferred by an electromagnetic wave depends on the size of a biological organism or, in general, the object that receives it. This phenomenon is called the *antenna factor* [6]. To account for the *antenna factor*, we used the specific intensity $P^*$ expressed as power P multiplied to the ratio between the size of the biological organism and the size of the detector.

The time for energy exchange depends on the frequency of a periodic phenomenon [5]. This happens because in periodic phenomena there is a strict connection between geometry, time and energy [5]. Specifically, with electromagnetic waves at low frequencies, such as radio waves, longer time intervals are required to transfer energy than with electromagnetic waves at higher frequencies, such as red and blue light. We argue that this time-dependence might cause a wavelength-dependence in many biological phenomena. To rule out underexposure, in our experiments we opted for long exposure times both with *C. elegans* (up to 1 hour) and with *G. jamesonii* (up to 192 hours).

## Video recording of C. elegans

To compare avoidance reactions of *C. elegans* to light and radio waves, we exposed about 40 nematodes to each one of the following environments: (1) white light, (2) blue light, (3) red light, and (4) radio waves. The experiments were repeated two to three times. The experimental setup is pictured in Fig 1. In environments (1), (2), and (3), the nematodes were kept

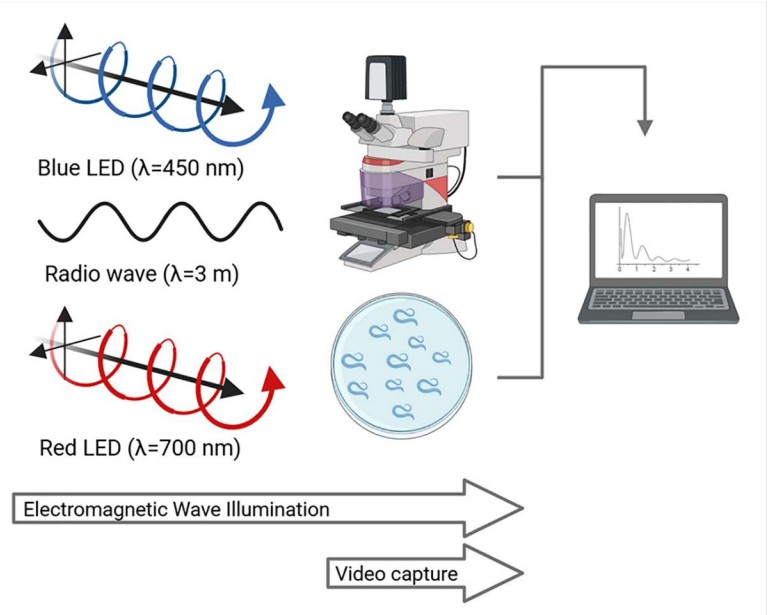

**Fig 1. Experimental setup used for imaging and analyzing *C. elegans* exposed to electromagnetic waves.** The setup involved the sources (blue light emitting diodes (LEDs), red LEDs, and radio wave generators), a plate with a sample of 40 freely moving *C. elegans* worms, a microscopy stage, and a computer for analysis. The symbol λ represents the wavelength of the electromagnetic waves. We pictured the blue and red light from the LEDs as helices to signify that light was not polarized. Radio waves were sketched as simple waves because their large wavelength (λ = 3 m) compared to the size of the nematodes makes it impossible to distinguish between polarized and unpolarized waves.

for 45 minutes in the dark and then exposed for 10 minutes to either white, blue or red light while video recording. In environment (4), nematodes were kept in darkness for 45 minutes while being exposed to radio waves, then the white light was turned on for 10 min to enable video recording. During these 10 minutes, *C. elegans* were exposed contemporarily to radio waves and white light. Videos were captured using a Leica M205 FCA Stereo Microscope equipped with a Leica DFC7000 GT camera. Exposures and video acquisition were conducted under similar conditions, including frame rate and resolution. All data for the *C. elegans* analyses are included in a GitHub repository at: https://github.com/Charlesbt12/C.-Elegans-EM-waves-RF.git.

## Photo recording of G. jamesonii

To quantify the petals' nyctinastic movements, we placed one flowering plant of *G. jamesonii* in each one of the following environments: (1) solar white light, (2) radio waves, (3) red light, (4) blue light, and (5) darkness. Starting the first day of blooming, in each environment we photographed the *G. jamesonii* petals of a single flower at 13:30 and then again at 16:30. We applied the same protocol on the following three days without removing the flowering plants from their environment. Examples of photographs of *G. jamesonii* in the five environments are shown in Fig 2. The photographs were taken with a Sony DSCWX350 18 MP Digital Camera. The number of flowers examined in each environment are as follows: 9 in solar white light, 5 in radio waves, 4 in red light, 3 in blue light, and 6 in darkness. The sample size was limited by logistics and time. The logistic limitation consisted of the fact that when we collected data we were able to process only three flowers per measurement set (one in dark and one in solar white light as controls, and one in either blue light, or in red light or in radio waves). The time limitation consisted of the fact that we could purchase *G. jamesonii* plants only in the flowering season which started in mid-May and ended by mid-August. Only recently, new experiments were performed by us with a

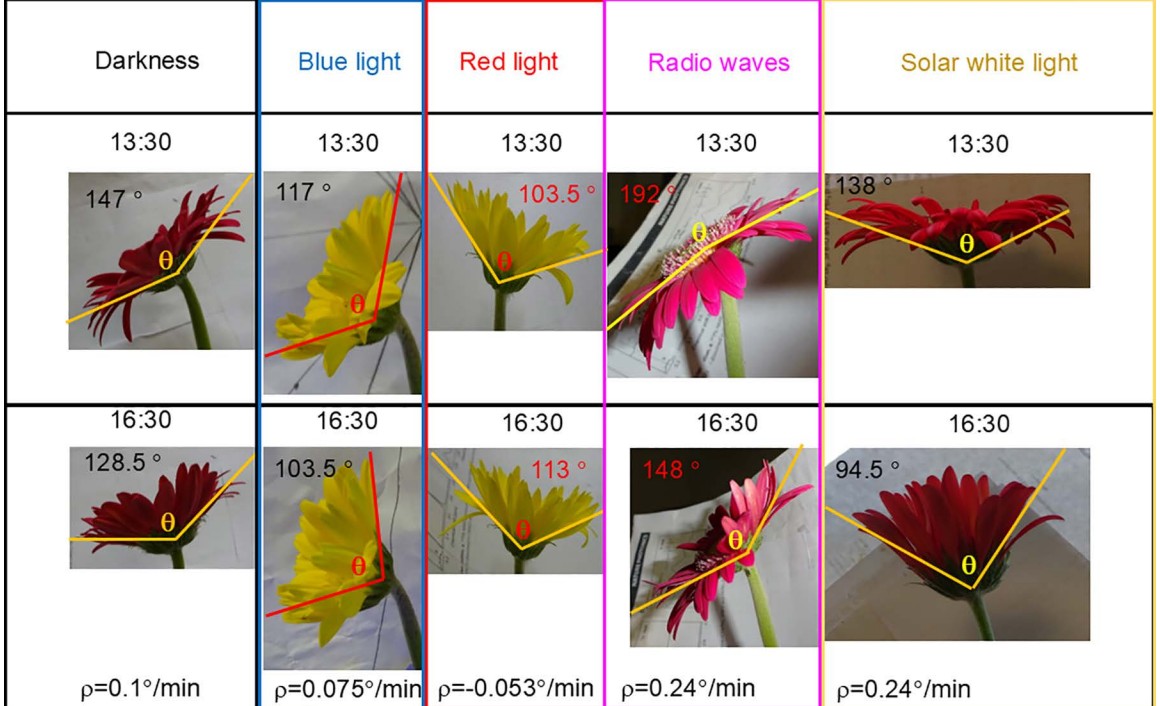

**Fig 2. Photographs of various *G. jamesonii*.** The images were taken on the first day of blossom at 13:30 (top) and 16:30 (bottom) in darkness, blue light, red light, radio waves, and solar white light. The aperture angles θ and the rate of closure $\rho = \Delta\theta/\Delta t^*$ are reported, where the time interval $\Delta t^*$ consists of the 180 minutes existing between 13:30 and 16:30.

different protocol and higher throughput (data not shown). The results strengthen those presented in this work and will be presented in future publication. If requested, we can provide a preview of these findings. Details on the experiments with *G. jamesonii* for this work are presented in S1 Protocol and S1 Dataset.

## Data analysis and statistical tests for *C. elegans*

We analyzed recorded videos using the Tierpsy Multi-Worm Behaviour Tracker [18], which extracts individual nematode tracks and associated kinematic and postural features. Each experimental condition consisted of four independent experiments, each containing 10 worms (biological N = 40 worms per condition). Tierpsy tracking generated thousands of locomotor tracks per plate over the recording period, representing repeated observations derived from individual worms (see https://github.com/Charlesbt12/C.-Elegans-EM-waves-RF.git and section 3 in S1 Text). Statistical analyses were performed using GraphPad Prism (version 10). Outliers were identified and removed using the Robust Outlier Identification (ROUT) method with maximum desired false discovery rate Q = 1%, and tracks with speeds exceeding 500 µm/s were excluded from further analysis. We used Welch's one-way ANOVA statistical test to compare absolute forward and reverse speeds between experimental groups, accounting for unequal variances. Mann-Whitney tests in a *post hoc* fashion were conducted to identify pairwise differences emerging among the results of the exposure of the worms in the various environments.

## Machine learning tests for *C. elegans*

To assess the relative importance of the response to electromagnetic waves of different postural and kinematic parameters of *C. elegans*, we employed a machine-learning based random forest classifier in R using RandomForest, caret,

and datasets packages [18–21], along with custom scripts available at https://github.com/Charlesbt12/C.-Elegans-EM-waves-RF.git. We input data collected through Tierpsy Multi-Worm Behaviour Tracker into the classifier and dropped the lowest possible number of incomplete observations, resulting in 282 observations, before classifying the observations collected in white light, blue light, red light, and radio waves. Measures of accuracy including the confusion matrices can be found in https://github.com/Charlesbt12/C.-Elegans-EM-waves-RF.git and in section 3 in S1 Text. We used the mean decrease Gini (MDG) index to determine the most important behavioral parameters based on their contribution to the accuracy of the random forest model to predict which electromagnetic waves were used in each recording.

### Angle measurement for the *G. jamesonii* petals

We photographed the aperture of the *G. jamesonii* petals and defined their angular aperture as θ. To measure θ, two lines were traced embracing the petals aperture and intersecting as close as possible to the point where the petals join the stem. This procedure is outlined in Fig 2. The morphology of the petal's aperture was found to be somewhat variable and did not allow us to establish universal rules to trace the lines for measuring θ. Therefore, we opted to tailor our measurements to each flower, first in taking the photographs and then in tracing the lines enabling the measurement of θ. We defined Δθ as the change in angular aperture of *G. jamesonii* during the time interval $\Delta t^* = 180$ min between 13:30 and 16:30 and the rate of closure as the ratio $\Delta\theta/\Delta t^*$. We consider the rate of closure as a measure of nyctinastic movements.

### Data analysis of the rate of closure ρ for *G. jamesonii*

For each illumination environment and for each of the four days of the experiment after blossoming started, we defined ρ as the average of the rate of closure measured in that specific environment for all the flowers we were able to test there. We defined the standard deviation of the mean (Δρ) to be the uncertainty on ρ. Due to the small sample size, a measurement of ρ was judged as statistically significant when the difference between two values of ρ was larger than their standard deviations. More details are provided in S1 Protocol and S1 Dataset.

## Results

### Radio waves and light influence the forward speed of *C. elegans*

Our initial microscopic examination of nematodes illuminated with blue light and radio waves revealed increased movement with fewer pauses and less stalling compared to those exposed to white light alone. Specific data about this finding were not recorded. These initial observations prompted us to analyze a staged young adult population of *C. elegans* using Tierpsy Multi-Worm Behaviour Tracker [18]. This approach enabled us to extract both event-based and time-series transformation features from plates containing multiple nematodes. A total of 3018 features were extracted and quantified from 20 videos for analysis with GraphPad Prism (version 10) software. Each of the 20 videos consisted of 298 worm tracks (extracted data are available in https://github.com/Charlesbt12/C.-Elegans-EM-waves-RF.git). We initially focused on the absolute values of forward and reverse velocities of *C. elegans* in each video and collected them in Fig 3. The average forward velocities of nematodes exposed to blue light (153.2 µm/s) and radio waves (91.7 µm/s) were notably higher than those exposed and video recorded under white light alone at 25% intensity (73.8 µm/s, with probability $p < 0.0001$). In contrast, *C. elegans* exposed to red light were marginally faster (83.96 µm/s, $p < 0.0001$) than those under white light. The absolute values of the reverse speeds were not strongly dependent upon the illumination environment ($p < 0.0001$). Altogether these findings suggested that particularly blue light and radio waves promoted rapid and consistent movements, signifying a response to noxious stimuli [8].

### Temperature negligibly affects the velocities of *C. elegans* in radio waves

We performed a dedicated experiment to understand the effect of temperature on the velocities of *C. elegans* in radio waves. From the examination of the data collected (Fig 4) we inferred that radio waves plus white light at 25% of the

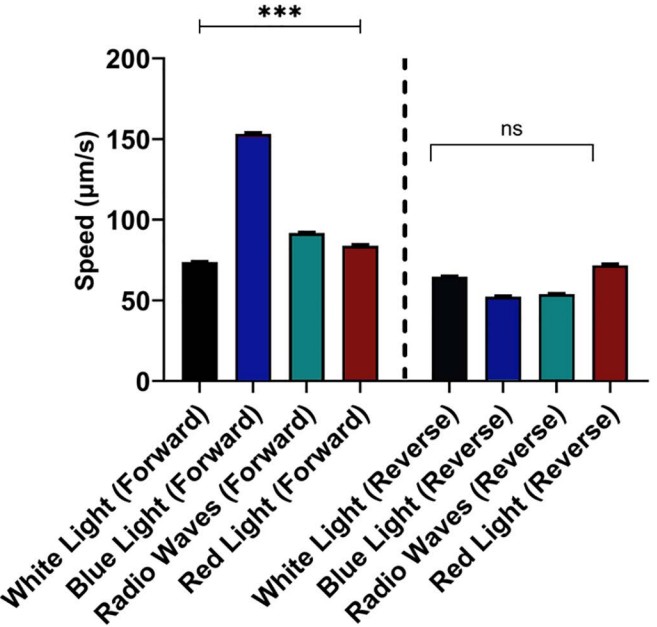

**Fig 3. At similar energies, blue light and radio waves augment the forward speed of *C. elegans*.** The average crawling speed (μm/s) was quantified for nematodes illuminated with either white light, blue light (λ = 450 nm), red light (λ = 700 nm), or radio waves (λ = 3 m). Nematodes exposed to blue light or radio waves exhibited significantly increased forward speeds compared to those under white light (***p < 0.0001, where p is probability), while speeds under red light were marginally faster than those under white light. Reverse locomotion speeds did not differ significantly (ns) across conditions. A dashed vertical line separates forward and reverse movement groups. Bars represent mean ± SEM (standard error of the mean). We tested 40 worms per treatment.

maximum intensity produced on *C. elegans* a smaller temperature increase (Fig 4a) and at a slower rate (Fig 4b) than in 100% white light alone. Thus, a 10-min exposure of *C. elegans* to radio waves plus white light at 25% intensity produced on average a 0.3 °C temperature increase. Such an increase in temperature heated the nematodes from about 23.3°C to about 23.6°C. We considered such temperature change negligible and concluded that heat neither affected the forward nor the reverse velocities of *C. elegans* exposed to radio waves [22].

### Temperature negligibly affects the rates of closure of *G. jamesonii*

We tested versus time the temperatures inside the EPS enclosures with *G. jamesonii* in all the environments we considered: darkness, blue light, red light and radio waves. Contemporarily, as controls, we measured the temperatures in the laboratory where the EPS enclosures were placed, and on the windowsill were *G. jamesonii* were exposed to solar white light. We reported the temperature versus time trends in Fig 4c. We started with blue light, red light and radio waves LEDs and radio devices turned on. At the 36th hour we turned LEDs and radio devices off and continued monitoring the temperatures. We observed that the temperature on the windowsill was up to 4°C larger than in the laboratory and in the EPS enclosures. In the enclosures, the temperatures remained almost constant with LEDs and radio devices on and off but followed the trends of the temperatures in the laboratory. We believe that these results attribute minimal influence of the environmental temperatures to the rates of closure of the *G. jamesonii* petals.

### Exposure to electromagnetic waves affects posture and locomotion of the entire *C. elegans* body

A machine learning based random forest classifier was built to identify the most distinguishing features among the 4083 recorded movement features. We summarized the results in Fig 5 upon analyzing the MDG index, from which we inferred

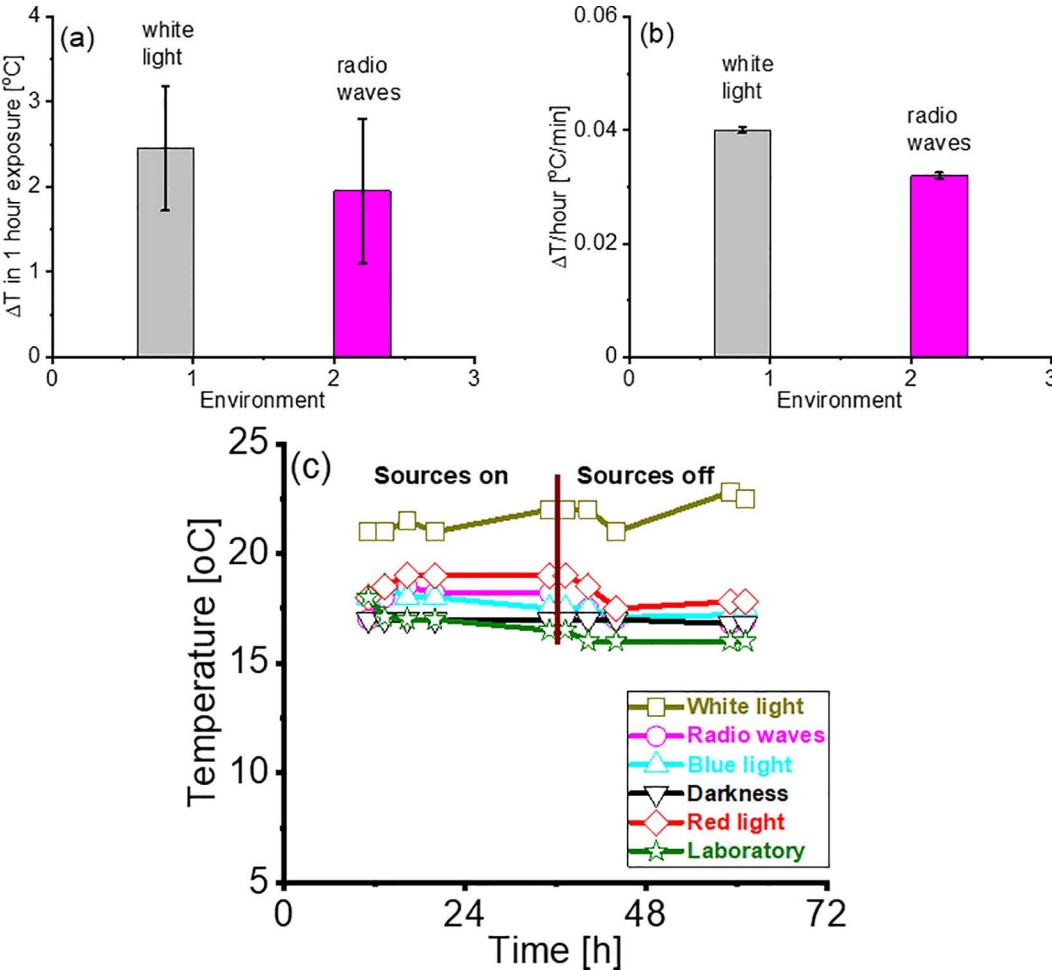

**Fig 4. Temperatures in our experiments negligibly affected *C. elegans* and *G. jamesonii*.** (a) Average temperature changes ΔT of *C. elegans* exposed to white light at maximum intensity (left bar) and to radio waves plus white light at reduced intensity (right bar) detected in one hour exposure; (b) corresponding rates of temperature increase. The average starting temperature was $(20.3 \pm 0.8)°C$. (c) Temperature versus time in the environments in which *G. jamesonii* were photo recorded. The vertical line at about the 36th hour corresponds to the point in time in which the sources (blue light, red light and radio waves) were turned off. The temperature in the laboratory is reported for reference.

three insights. One was that the parameters most significantly contributing to the accuracy of the random forest model for a specific environment were [18]: B, the body shape, which defines the nematode's posture and its variations; R, the rate of postural change; ω, the radial speed or the movement of the *C. elegans* head relative to its body; finally, S, the forward speed, which reveals how fast a nematode proceeds headfirst. The other insight was that the B, R, S and ω parameters correlated to speed. Thus, the small differences in forward speed of *C. elegans* exposed to white light, red light, and radio waves seen in Fig 3 were due to differences in speeds, not to other factors such as recording problems or error propagation. The third insight that emerged from the data presented in Fig 5 was the appearance of parameters B and R among the most prominent ones. This finding suggested that exposure to all the tested illumination environments induced changes in posture, including fine adjustments, that were more subtle and complex than simple alterations in movement or basic locomotion [21] affecting the coordination of the entire *C. elegans* body posture.

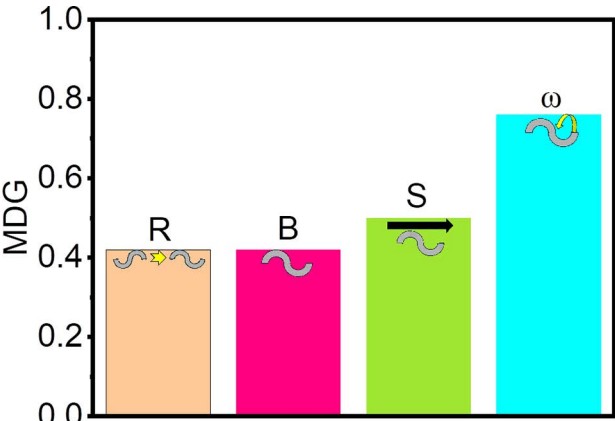

**Fig 5. *C. elegans* behavioral parameters are those most affected by electromagnetic waves of similar energies.** Mean Decrease Gini (MDG) index for the four most important parameters: the body shape B, rate of posture change R, forward speed S and radial speed ω, singled out in the analysis of populations of *C. elegans* in red light, blue light, white light, and radio waves. We used the MDG index to determine the most important behavioral parameters based on their contribution to the accuracy of the random forest model to predict which electromagnetic waves were used in each recording.

### Radio waves and light influence the rate of closure ρ of *G. jamesonii*

We photographed the aperture of the petals of *G. jamesonii* at 13:30 and at 16:30 in all four days of each experiment to capture the angular aperture θ. Selected photographs are shown in Fig 2 (more details are presented in S1 Dataset.). From the photographs taken in each illumination environment and for each of the four days of the experiments, we measured the rate of closure ρ of *G. jamesonii* and its uncertainty Δρ. The results were presented in Fig 6. In darkness, ρ did not significantly change in the four days of the experiment, as variations in ρ were smaller than Δρ. In blue light, ρ presented the same trend as in darkness. In red light, ρ underwent small variations but, differently than in darkness and blue light, the petals of *G. jamesonii* opened in the time interval between 13:30 and 16:30. This phenomenon resembled that induced by red light in the pinnulae of *Mimosa pudica* (*M. pudica*) and observed by S. Setty *et al*. [23]. In radio waves, the petals of *G. jamesonii* showed a remarkable behavior: first, ρ was larger than in darkness, blue light, and red light; second, variation in ρ over the four days of the experiment was statistically significant; third, the petals nyctinastic movement transitioned from closing in days 1 and 2 to opening in days 3 and 4. Finally, in solar white light, ρ was significantly larger than in any of the other examined illumination environments on each day of the experiments, and the nyctinastic movements were sharp. Moreover, ρ exhibited a gentle decrease from the first to the fourth day since blossom started. We considered the trend of ρ in solar white light as representative of the circadian cycle in *G. jamesonii*. [24].

### Discussion

Numerous studies have investigated the effects of electromagnetic waves on biological organisms. Recently, radio waves and microwaves received significant attention due to their use in communication systems such as telephones, television, radio transmission, and telecommunication on Earth and in space. However, there is no consensus regarding how to establish when radio waves and microwaves become harmful on biological organisms. We hypothesized that the contradictory results published in the current literature stemmed from the lack of an adequate method to evaluate the energy transferred from electromagnetic waves to biological organisms and the intensity needed to achieve such energy. To overcome this problem, we tested the recently proposed classical expression of the electromagnetic wave energy (i.e., the product of intensity with the inverse of the wave's frequency) [5,6] as a measure of the energy transferred to biological organisms. We selected *C. elegans* and *G. jamesonii* as representative biological organisms to be exposed to radio

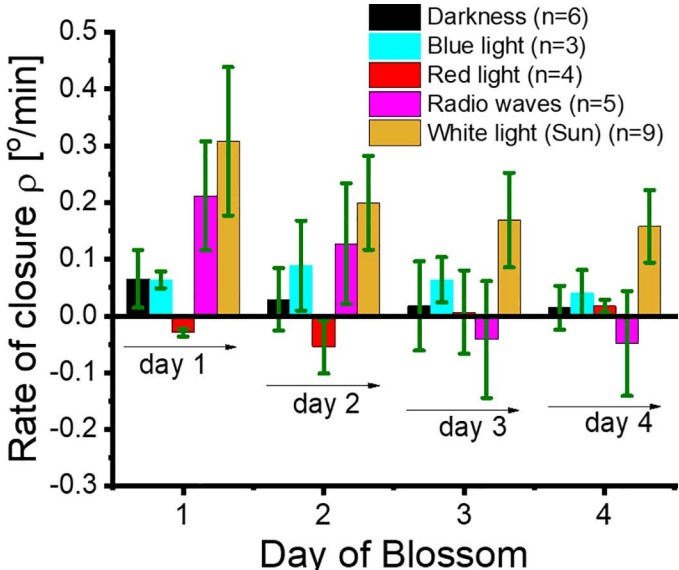

**Fig 6. Radio waves significantly affect the rate of closure ρ of *G. jamesonii*.** The trends of ρ are presented here versus day of blossom for *G. jamesonii* in various environments. The environments considered were darkness, blue light, red light, radio waves, and solar white light. The rate of closure was measured each day between 13:30 and 16:30, the photo period in which *G. jamesonii* in solar white light experiences closure at the fastest rate. The data collection started mid-May up to mid-August. The bars represent the average value of ρ. The error bars are the standard deviations Δρ. n represents the number of flowers used to measure ρ in each environment.

waves and light and on which to measure the amount of energy received. The rationale of our procedure was that, if radio waves and light transfer a similar amount of energy (within about one order of magnitude) to the biological organisms, they should trigger similar responses.

We found that light and radio waves affected the forward escape velocities of *C. elegans* to a similar degree. The reverse velocities were influenced to a significantly lesser degree, and we neglected further investigation. Using machine learning-based tools, we confirmed that the differences in the forward escape velocities generated by blue light, red light, and radio waves arose from the kinematic and postural parameters of *C. elegans*, not from recording problems or error propagation. We also assessed that temperature changes were negligible and did not affect our findings. We ascribed the similar magnitude of the forward escape velocities to the fact that the classic electromagnetic wave energy of the blue light, red light, and radio waves used in our experiments were within one order of magnitude of each other, as we showed in Table 1.

The daily cycle of darkness and brightness of solar white light gives rise to the periodic opening and closing of the *G. jamesonii* petals, which is a phenomenon easy to observe, even with naked eyes. During the flowering season, in solar white light *G. jamesonii* petal closure is sharp in the day's photoperiod between 13:30–16:30. In our experiments, we found that continuous exposure to monochromatic light (such as blue and red light) kept at the same energy inhibited the cyclic nyctinastic movement of the *G. jamesonii* petals. We also found that, in the targeted photo period, red light promoted petal opening, as opposed to blue light, which triggered slow petal closure. This behavior is in line with that observed in previous studies which attributed *G. jamesonii* petals' response to red light to the excitation of different phytohormones [11]. Qingqing Wang *et al*. [25] associated the peculiar effects of red light on *Arabidopsis thaliana* to a shift in photoperiod, a phenomenon that could pertain to *G. jamesonii* as well. What surprised us was the discovery that petal closure of *G. jamesonii* continuously exposed to radio waves was not only observable, but also that it occurred at a rate like that in solar white light, with ρ in radio waves being slightly lower than in solar white light (Fig 6). We ascribed the

differences in ρ of *G. jamesonii* exposed to radio waves compared with that in blue light and red light to the fact that the classic electromagnetic wave energy of the radio waves we used in our experiments was about one order of magnitude larger than that of the red light and blue light, shown in Table 2. More investigation is needed to clarify this phenomenon.

We pondered mechanisms alternative to energy that could explain our results. Our data did not detect significant temperature changes in our biological organisms and in the environments where we tested them. Thus, we ruled out temperature variations to be an alternative mechanism. Neither noise nor sound were present in the environments in which we exposed our biological organisms. Thus, we ruled out vibration-related effects as an alternative mechanism. We performed our experiments in EPS enclosures, which we selected for their shielding capability thus enabling us to rule out external interferences and disturbances as alternative mechanisms. Only the tests of *G. jamesonii* in solar white light were performed without shielding. However, in that environment our findings singled out the behavior of the flower in its natural condition. Finally, since we exposed our biological organisms for long time intervals (up to 1 hour for *C. elegans* and 192 hours for *G. jamesonii*) and gave them sufficient time to interact with both light and radio waves, we ruled out under- or over-exposure to electromagnetic waves to be alternative mechanisms. This conclusion is further supported by the fact that, from the point of view of the electromagnetic wave energy, the amount of energy exchanged depends on power and frequency, not on exposure duration [6]. We could argue that cellular mechanisms might be at the origin of our findings, however further investigations are needed to elucidate this possibility. From these considerations we concluded that the fact that the electromagnetic wave energy provided similar energy in the tested environments on our biological organisms remains the most probable explanation.

Altogether, our findings confirmed known or expected behaviors of *C. elegans* and *G. jamesonii* in blue and red light. In addition, our findings suggested that radio waves with classical electromagnetic wave energy within an order of magnitude of that of light trigger similar behavioral trends to those normally triggered by light. Based on this result, we propose that the classic electromagnetic wave energy [5], i.e., the product of wave's intensity with the inverse of its frequency, correctly accounts for the energy transferred to biological organisms. The cellular mechanisms involved in the interaction between radio waves and either *C. elegans* or *G. jamesonii* need to be further investigated.

Having assessed that the classic electromagnetic wave energy [5,6], i.e., the product of wave's intensity with the inverse of its frequency, accounts for the energy transferred from electromagnetic waves to biological organisms, we were able to draw conclusions regarding the intensity. Since intensity and frequency are directly related, electromagnetic waves at large frequency (e.g., light in the $10^{15}$ Hz range) have larger intensities than radio waves, which have lower frequencies, between $10^6$ Hz and $10^9$ Hz. Indeed, we showed in Tables 1 and 2 that our radio wave intensities were in the $10^{-9}$ W (nW) range while our light intensities were in the $10^{-3}$ W (mW) range. The intensities expressed in W m$^{-2}$ Tables 1 and 2 followed similar trends. This observation constitutes a criterion enabling the choice of the magnitude of the intensity of radio waves and microwaves and highlighting how intensity, frequency, and duration of exposure are related to the energy transferred from the waves to a biological organism.

As an application of our study, we envision that radio waves with proper intensity and frequency could activate radio-genetic processes as a non-invasive substitute of blue light-based optogenetics [26]. The advantage of radio waves over blue light is their larger penetration depth into the tissues of biological organisms, promising to make radio-genetics an attractive, non-invasive diagnostics and therapy tool.

## Conclusions

We observed that *C. elegans* nematodes responded to radio waves by increasing their forward speed in a manner that resembles the response to blue light. We also observed that *G. jamesonii* petals close their petals at a rate slightly lower than that under solar white light and higher than that under blue and red light. We thus inferred that the classic electromagnetic wave energy given as the product between intensity and inverse of the frequency described the amount of energy transferred from electromagnetic waves to biological organisms. We showed that the classic electromagnetic wave

energy established a criterion to select the wave intensity once the amount of energy to be transferred on a biological organism was specified. We noted that, in transferring the same amount of energy, radio waves required a significantly lower intensity than light. Further studies based on these results could lead to the development of radio-genetics as a non-invasive alternative to optogenetics in the diagnosis and treatment of neurological pathologies.

## Supporting information

**S1 Protocol. Protocol for experiments with *G. jamesonii*.** Summary of the procedure we used to obtain the results on *G. jamesonii* presented in this work.
(DOCX)

**S1 Dataset. Images and data for the analysis of *G. jamesonii*.** This file contains tables and images with the raw data that enabled extracting the results for the rate of closure of *G. jamesonii* presented in this work.
(PDF)

**S1 Text. Additional text.** This file contains additional information on the shielding effect of the EPS enclosures, on the classic electromagnetic waves energy in scientific literature, and on the accuracy of the Random Forest Classifier.
(DOCX)

## Acknowledgments

We thank James A. Clabough for contributing to the experiments in the early stages of the research.

## Author contributions

**Conceptualization:** Timothy A. Bloss, Marquis T. Walker, Kristopher L. Schmidt, Giovanna Scarel.

**Data curation:** Alexander W. Kline, Charles S. Beattie, Addison K. Shenk, Samuel I. Spicher, Timothy A. Bloss, Laura G. Vallier, Kristopher L. Schmidt, Giovanna Scarel.

**Formal analysis:** Alexander W. Kline, Charles S. Beattie, Addison K. Shenk, Samuel I. Spicher, Timothy A. Bloss, Laura Tipton, Kristopher L. Schmidt, Giovanna Scarel.

**Funding acquisition:** Timothy A. Bloss, Marquis T. Walker, Kristopher L. Schmidt, Giovanna Scarel.

**Investigation:** Alexander W. Kline, Addison K. Shenk, Samuel I. Spicher, Timothy A. Bloss, Laura Tipton, Kristopher L. Schmidt, Giovanna Scarel.

**Methodology:** Timothy A. Bloss, Marquis T. Walker, Kristopher L. Schmidt, Giovanna Scarel.

**Project administration:** Giovanna Scarel.

**Resources:** Timothy A. Bloss, Giovanna Scarel.

**Software:** Laura Tipton, Kristopher L. Schmidt.

**Supervision:** Timothy A. Bloss, Laura Tipton, Marquis T. Walker, Kristopher L. Schmidt, Giovanna Scarel.

**Validation:** Alexander W. Kline, Charles S. Beattie, Addison K. Shenk, Samuel I. Spicher, Timothy A. Bloss, Laura Tipton, Marquis T. Walker, Laura G. Vallier, Kristopher L. Schmidt, Giovanna Scarel.

**Visualization:** Laura Tipton.

**Writing – original draft:** Laura G. Vallier, Kristopher L. Schmidt, Giovanna Scarel.

**Writing – review & editing:** Alexander W. Kline, Charles S. Beattie, Addison K. Shenk, Samuel I. Spicher, Timothy A. Bloss, Laura Tipton, Marquis T. Walker, Laura G. Vallier, Kristopher L. Schmidt, Giovanna Scarel.

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
