## [Decision Letter · Decision Letter 0]

23 Dec 2025

Dear Dr. Scarel,

Thank you for submitting your manuscript to PLOS ONE. After careful consideration, we feel that it has merit but does not fully meet PLOS ONE’s publication criteria as it currently stands. Therefore, we invite you to submit a revised version of the manuscript that addresses the points raised during the review process.

We look forward to receiving your revised manuscript.

Kind regards,

Satish kumar Rajasekharan

Academic Editor

PLOS One

Journal Requirements:

“We acknowledge NSF Major Research Instrumentation Grant No. 1827997 and 4-VA JMU for funding; the Department of Physics and Astronomy at JMU, the Department of Biology at JMU, and the Department of Biology & Chemistry at EMU for support.  CSB acknowledges a Tickle Scholarship in Summer 2025 at JMU.  We thank Dr. Laura G. Vallier for reviewing and editing the manuscript, and James A. Clabough contributing to the experiments in the early stages of the research.”

“NSF Major Research Instrumentation Grant No. 1827997

4-VA JMU”

“NSF Major Research Instrumentation Grant No. 1827997

4-VA JMU”

Reviewers' comments:

Reviewer's Responses to Questions

**Comments to the Author**

1. Is the manuscript technically sound, and do the data support the conclusions?

Reviewer #1: Yes

Reviewer #2: Yes

2. Has the statistical analysis been performed appropriately and rigorously?

Reviewer #1: No

Reviewer #2: Yes

3. Have the authors made all data underlying the findings in their manuscript fully available?

Reviewer #1: Yes

Reviewer #2: Yes

4. Is the manuscript presented in an intelligible fashion and written in standard English?

Reviewer #1: Yes

Reviewer #2: Yes

Reviewer #1: The use of both C. elegans and G. jamesonii as model systems is innovative. However, the manuscript has several major scientific and methodological weaknesses, including insufficient controls, incomplete data interpretation, and limited statistical rigor. The proposed “classical electromagnetic wave energy” concept lacks theoretical justification and empirical validation. Extensive revision and additional experimentation are needed before this work can be considered for publication in PLOS ONE.

The definition of “classic electromagnetic wave energy” as P/ν is nonstandard and not theoretically derived from Maxwellian or quantum electrodynamics principles. Authors must justify this equation with clear theoretical or empirical grounding.

The study assumes that equal P/ν values between radio waves and visible light correspond to equal biological impact. This assumption oversimplifies photon–matter interactions and ignores wavelength-dependent absorption and biological chromophore sensitivity.

The rationale for selecting specific intensity and frequency combinations to equate “energy” between blue/red light and radio waves is unclear. Provide calculations showing that biological exposure levels are physically comparable and within realistic dose-response ranges.

The number of replicates for both C. elegans (n=10 worms per group) and G. jamesonii (3–9 flowers per treatment) is insufficient for robust statistical inference. Power analysis and justification for sample size are required.

Missing controls such as exposure to “radio-off” or shielded conditions should be added to distinguish electromagnetic effects from environmental or mechanical stimuli.

Although temperature was measured for C. elegans, no similar thermal monitoring was reported for G. jamesonii. Continuous illumination may have altered local microclimate conditions. This must be evaluated and controlled.

The random forest classifier is used but not validated. Include model accuracy metrics, confusion matrices, and justification for parameter selection. Explain how overfitting was avoided.

The authors should discuss alternative mechanisms such as weak thermal gradients, vibration, or electromagnetic interference effects rather than implying direct photochemical or radio-genetic activation.

The authors should cite the article “LED-based real-time survival bioassays for nematode research” because it presents validated LED illumination systems and quantitative behavioral assays for C. elegans. Citing it would strengthen the methodological justification for using LED-based illumination and standardize behavioral recording conditions. It provides a well-established LED-driven bioassay platform relevant to the current study’s illumination setup and supports the use of LEDs as biologically meaningful light sources for nematode behavioral experiments.

Provide explicit details on the number of biological replicates, duration of experiments, and light intensities at organism level (W/m²).

Reviewer #2: The research on the influence of intensity and frequency of electromagnetic waves on the biological system has been conducted with scientific rigour. Evaluations were carried out extensively justifying the hypothesis. Though the methods, results, and discussion were well documented,

Authors are suggested to emphasize the findings, that lead to the justification of hypothesis, in the conclusion session.

**Do you want your identity to be public for this peer review?** For information about this choice, including consent withdrawal, please see our For information about this choice, including consent withdrawal, please see our Privacy Policy .

Reviewer #1: No

Reviewer #2: No

---

## [Author Response · Author response to Decision Letter 1]

2 Feb 2026

Responses to the Editor:

Editor’s comment: If applicable, we recommend that you deposit your laboratory protocols in protocols.io to enhance the reproducibility of your results. Protocols.io assigns your protocol its own identifier (DOI) so that it can be cited independently in the future. For instructions see: https://journals.plos.org/plosone/s/submission-guidelines#loc-laboratory-protocols.

Response to Editor: The procedure for preparing C. elegans is well established and as reported in Refs. 9 and 10 in the Main Text. Thus, we believe not to be necessary to outline the protocol. For G. jamesonii we created a S1 Protocol document that summarizes step-by-step how we achieved the presented results. We prefer to defer to a later time to put our procedure in protocols.io.

Editor’s comment: Please ensure that your manuscript meets PLOS ONE's style requirements, including those for file naming. The PLOS ONE style templates can be found at https://journals.plos.org/plosone/s/file?id=wjVg/PLOSOne_formatting_sample_main_body.pdf and https://journals.plos.org/plosone/s/file?id=ba62/PLOSOne_formatting_sample_title_authors_affiliations.pdf

Response to Editor: We did as we were requested. Please let us know if we forgot anything.

Editor’s comment: Please remove any funding-related text from the manuscript and let us know how you would like to update your Funding Statement. Please include your amended statements within your cover letter; we will change the online submission form on your behalf.

Response to Editor: Here are our funding sources:

a) NSF Major Research Instrumentation Grant No. 1827997,

b) 4-VA JMU Testing the efficacy of radio waves to activate biological receptors (2021),

c) Department of Physics and Astronomy JMU, Department of Biology JMU, and Department of Biology & Chemistry EMU for support,

d) Jeffrey E. Tickle '90 Family Endowment in Science & Mathematics Scholarship, College of Science and Mathematics JMU,

e) Physics and Astronomy Nadine Barlow Undergraduate Research Support Award 2026 from the Council of Undergraduate Research (CUR), (Pending).

Editor’s comment: Please state what role the funders took in the study.

Response to Editor: Here are our responses on the role of the Funders:

a) NSF Major Research Instrumentation Grant No. 1827997: provided funding the Leica M205 FCA Stereo Microscope equipped with a Leica DFC7000 GT camera.

b) 4-VA JMU Testing the efficacy of radio waves to activate biological receptors (2021): provided funding for materials and supplies and for a student (James A. Clabough) at the start of the project,

c) the Department of Physics and Astronomy JMU, the Department of Biology JMU, and the Department of Biology & Chemistry EMU: supported A.W.K in Summer 2023 and C.S.B in Summer 2024,

d) the Jeffrey E. Tickle '90 Family Endowment in Science & Mathematics Scholarship supported C.S.B., Summer 2025,

e) the 2026 Nadine Barlow Undergraduate Research Support Award, if awarded will contribute to partially cover the publication fee for this manuscript.

Editor’s Comment: We note that your Data Availability Statement is currently as follows: All relevant data are within the manuscript and its Supporting Information files.

Please confirm at this time whether your submission contains all raw data required to replicate the results of your study. Authors must share the “minimal data set” for their submission. PLOS defines the minimal data set to consist of the data required to replicate all study findings reported in the article, as well as related metadata and methods (https://journals.plos.org/plosone/s/data-availability#loc-minimal-data-set-definition).

- The values behind the means, standard deviations and other measures reported.

Response to Editor: All data for the C. elegans analyses are now included in the GitHub repository linked on page 15 of the revised manuscript: https://github.com/Charlesbt12/C.-Elegans-EM-waves-RF.git. On page 10 of the revised manuscript, we link another GitHub repository containing files describing step-by step how the classic electromagnetic wave energy is derived from Maxwell’s equations in the case of a simple oscillating dipole. All data for the G. jamesonii analyses are reported in S1 Dataset. We have updated the Supporting Information section on page 31 of the revised manuscript. We chose the GitHub repository because of its compatibility with the GitHub page of Tierpsy Multi-Worm Behavior Tracker, easy shareability and free accessibility. If requested by the Editor, however, we can change repository.

Editor’s Comment: If the reviewer comments include a recommendation to cite specific previously published works, please review and evaluate these publications to determine whether they are relevant and should be cited. There is no requirement to cite these works unless the editor has indicated otherwise.

Response to Editor: Reviewer # 1 recommended citing an article now referenced it in our manuscript (Ref. 12 in the revised manuscript).

Editor’s Comment: To ensure your figures meet our technical requirements, please review our figure guidelines: https://journals.plos.org/plosone/s/figures. You may also use PLOS’s free figure tool, NAAS, to help you prepare publication quality figures: https://journals.plos.org/plosone/s/figures#loc-tools-for-figure-preparation. NAAS will assess whether your figures meet our technical requirements by comparing each figure against our figure specifications.

Response to Editor: We believe we comply with the provided guidelines. All our figures are submitted in .tiff.

Responses to the Reviewers:

Reviewer #1: The use of both C. elegans and G. jamesonii as model systems is innovative. However, the manuscript has several major scientific and methodological weaknesses, including insufficient controls, incomplete data interpretation, and limited statistical rigor. The proposed “classical electromagnetic wave energy” concept lacks theoretical justification and empirical validation. Extensive revision and additional experimentation are needed before this work can be considered for publication in PLOS ONE.

Reviewer #1 Comment n. 1: The definition of “classic electromagnetic wave energy” as P/ν is nonstandard and not theoretically derived from Maxwellian or quantum electrodynamics principles. Authors must justify this equation with clear theoretical or empirical grounding.

Response: Although not commonly used, the classic electromagnetic wave energy is well rooted in classical and relativistic electrodynamics. In some simple cases, such as that of an oscillating dipole, it is possible to derive the classic electromagnetic wave energy in a simple way directly from Maxwell’s equations. In more complex cases it is possible to show that the classic electromagnetic wave energy accounts for the energy involved and measured experimentally by other means. Here we add that often the electromagnetic wave energy is designated as P*� instead of P/�, where P is power, � is frequency and � is period and �=1/�. We decided to choose P/� to limit the number of variables to be defined and simplify the text. We addressed the Reviewer’s comment by adding the new Section within the manuscript entitled “The classic electromagnetic wave energy” on pages 9-12 of the revised manuscript and in section 2 of S1 Text.

Reviewer #1 Comment n. 2: The study assumes that equal P/ν values between radio waves and visible light correspond to equal biological impact. This assumption oversimplifies photon–matter interactions and ignores wavelength-dependent absorption and biological chromophore sensitivity.

Response: We addressed this comment in the new Section within the manuscript entitled “The classic electromagnetic wave energy” on page 12, third line from the top.

Reviewer #1 Comment n. 3: The rationale for selecting specific intensity and frequency combinations to equate “energy” between blue/red light and radio waves is unclear. Provide calculations showing that biological exposure levels are physically comparable and within realistic dose-response ranges.

Response: We addressed this comment and provided the calculations required in the new Section within the manuscript entitled “The classic electromagnetic wave energy” on pages 11-12, starting from the paragraph under Eq. (4).

Reviewer #1 Comment n. 4: The number of replicates for both C. elegans (n=10 worms per group) and G. jamesonii (3–9 flowers per treatment) is insufficient for robust statistical inference. Power analysis and justification for sample size are required.

Response: A post-hoc power analysis is not included as this is not best statistical practice, however, we have added additional exact p-values to the manuscript to highlight that these findings are more likely to be the result of true differences than a lack of power (page 18-text in red). For the C. elegans, we have been more explicit about the sample sizes used as there were approximately 40 worms per plate and multiple tracks recorded per plate (see page 15-16-text in red). For G. jamesonii: we explained the rationale of the sample size and the number of replicates in the revised manuscript on page 14-text in red- and in S1 Protocol on page 5.

Reviewer #1 Comment n. 5: Missing controls such as exposure to “radio-off” or shielded conditions should be added to distinguish electromagnetic effects from environmental or mechanical stimuli.

Response: For C. elegans, it was not possible to perform experiments in darkness because lack of light does not allow video recording. Imaging C. elegans in an environment without radio waves requires light and thus would not work as a control. For G. jamesonii the measurements performed in darkness, with neither radio waves nor lights, acted as controls. We highlight the role of measurements in darkness on page 6-last line- of the revised manuscript. Moreover, on page 7- text in red- we highlighted the need to perform our experiments in shielded environments because without shielding the radio waves are dispersed and disturbed. We further discussed this issue in section 1 of S1 Text.

Reviewer #1 Comment n. 6: Although temperature was measured for C. elegans, no similar thermal monitoring was reported for G. jamesonii. Continuous illumination may have altered local microclimate conditions. This must be evaluated and controlled.

Response: We performed new measurements to address this comment and reported the results in the new section “Temperature negligibly affects the rates of closure of G. jamesonii” on page 20-21- text in red- and in the updated Fig. 4 of the revised manuscript.

Reviewer #1 Comment n. 7: The random forest classifier is used but not validated. Include model accuracy metrics, confusion matrices, and justification for parameter selection. Explain how overfitting was avoided.

Response: The random forest classifier was used for feature selection rather than classification. We have made this more explicit on page 21-22 (Section on “Exposure ……” of the revised manuscript but have added the accuracy metrics and confusion matrices to Table S3 in S1 Text. Since classification and generalizability were not the goal, we did not attempt to avoid overfitting beyond splitting the data into training and testing data with a 70/30 split.

Reviewer #1 Comment n. 8: The authors should discuss alternative mechanisms such as weak thermal gradients, vibration, or electromagnetic interference effects rather than implying direct photochemical or radio-genetic activation.

Response: We addressed this comment in the Discussion of the revised manuscript on page 26-27-text in red.

Reviewer #1 Comment n. 9: The authors should cite the article “LED-based real-time survival bioassays for nematode research” because it presents validated LED illumination systems and quantitative behavioral assays for C. elegans. Citing it would strengthen the methodological justification for using LED-based illumination and standardize behavioral recording conditions. It provides a well-established LED-driven bioassay platform relevant to the current study’s illumination setup and supports the use of LEDs as biologically meaningful light sources for nematode behavioral experiments.

Response: We thank the Reviewer for the suggestion and did accordingly by adding the new reference 12 and new text on page 8 of the revised manuscript (second line from the top).

Reviewer #1 Comment n. 10: Provide explicit details on the number of biological replicates, duration of experiments, and light intensities at organism level (W/m²).

Response: For C. elegans the requested details are reported on pages 17-18-text in red- of the revised manuscript (text in red at the bottom of page 17 and following). For G. jamesonii the details on the number of biological replicates and duration of experiments are reported on pages 13-14-text in red- of the revised manuscript, and in S1 Protocol. The intensities at the organismal level in W/m² estimated as outlined on page 8- text in red, five lines above the bottom) are reported in the updated Tables 1 and 2 of the revised manuscript. As we pointed out on page 27-last line, intensities in W/m² inferred the same trends as those we already found.

Reviewer #2: The research on the influence of intensity and frequency of electromagnetic waves on the biological system has been conducted with scientific rigor. Evaluations were carried out extensively justifying the hypothesis. Though the methods, results, and discussion were well documented.

Reviewer #2 Comment n. 1: Authors are suggested to emphasize the findings that lead to the justification of hypothesis in the conclusion session.

Response: We thank the Reviewer for the suggestion and accordingly updated the Conclusions on page 28-text in red- of the revised manuscript.

We hope that our responses clarify the concerns raised by Editor and Reviewers. Please let us know if any response is not complete.

Thank you in advance for your consideration.

Sincerely,

Giovanna Scarel, Kristopher Schmidt, Alexander W. Kline, Charles S. Beatti, Addison K. Shenk, Samuel I. Spicher, Timothy A. Bloss, Laura Tipton, Marquis T. Walker, Laura G. Vallier.

---

## [Editor Report · Decision Letter 1]

8 Feb 2026

A phenomenological comparison of the effects of blue light, red light and radio waves on the escape speed of Caenorhabditis elegans and the rate of closure of Gerbera jamesonii petals

PONE-D-25-47579R1

Dear Dr. Scarel,

We’re pleased to inform you that your manuscript has been judged scientifically suitable for publication and will be formally accepted for publication once it meets all outstanding technical requirements.

Kind regards,

Satish kumar Rajasekharan

Academic Editor

PLOS One

---

## [Editor Report · Acceptance letter]

PONE-D-25-47579R1

PLOS One

Dear Dr. Scarel,

I'm pleased to inform you that your manuscript has been deemed suitable for publication in PLOS One. Congratulations! Your manuscript is now being handed over to our production team.

Kind regards,

on behalf of

Dr. Satish kumar Rajasekharan

Academic Editor

PLOS One